# How Does Added Mass Affect the Gait of Middle-Aged Adults? An Assessment Using Statistical Parametric Mapping

**DOI:** 10.3390/s22166154

**Published:** 2022-08-17

**Authors:** Vinayak Vijayan, Shanpu Fang, Timothy Reissman, Megan E. Reissman, Allison L. Kinney

**Affiliations:** Department of Mechanical and Aerospace Engineering, University of Dayton, Dayton, OH 45410, USA

**Keywords:** added mass, walking, middle-aged, exoskeletons, statistical parametric mapping, biomechanics, kinematics, kinetics, EMG, muscle activity

## Abstract

To improve exoskeleton designs, it is crucial to understand the effects of the placement of such added mass on a broad spectrum of users. Most prior studies on the effects of added mass on gait have analyzed young adults using discrete point analysis. This study quantifies the changes in gait characteristics of young and middle-aged adults in response to added mass across the whole gait cycle using statistical parametric mapping. Fourteen middle-aged and fourteen younger adults walked during 60 s treadmill trials under nine different loading conditions. The conditions represented full-factorial combinations of low (+3.6 lb), medium (+5.4 lb), and high (+10.8 lb) mass amounts at the thighs and pelvis. Joint kinematics, kinetics and muscle activations were evaluated. The young and middle-aged adults had different responses to added mass. Under pelvis loading, middle-aged adults did not adopt the same kinematic responses as younger adults. With thigh loading, middle-aged adults generally increased knee joint muscle activity around heel strike, which could have a negative impact on joint loading. Overall, as age may impact the user’s response to an exoskeleton, designers should aim to include sensors to directly monitor user response and adaptive control approaches that account for these differences.

## 1. Introduction

In general, lower limb exoskeletons with sensor-guided (user-volitional) control strategies are commonly designed to provide assistance at either the hip or the ankle joints [1]. Different exoskeleton control strategies exist, with the most common strategies employing either motion or torque-based control [2,3]. Torque-based control is the most common method for providing partial assistance to a user to improve performance during movement [3] and could be applied to compensate for changes in gait due to aging [4]. In comparison to ankle exoskeletons, hip exoskeletons add only a minimal amount of inertia to the lower extremities and are therefore thought to minimally impact the resultant lower body dynamics during walking [1]. It has been previously noted in the literature that users who may benefit from using an exoskeleton to improve mobility might not choose to wear an exoskeleton due to disadvantages of the design, including the effort required for the donning and doffing of an exoskeleton [4]. More user-friendly exoskeletons that improve accessibility are a suggested focus for future exoskeleton designers [4]. One step towards designing such a user-friendly exoskeleton is fully quantifying the effects of placing added mass in the form of an exoskeleton on a broad spectrum of users, and especially in middle-aged or older adults who are likely to benefit from exoskeleton use. A one-size-fits-all approach to the design of exoskeletons will impede access to safe and effective technology for a large part of the population. In a study about the perception of exoskeleton technology, all the older adults surveyed were not aware of exoskeletons as a technology, and they did not perceive the need for an exoskeleton [5]. It has also been shown that older adults are less likely to adopt the use of technology [6]. Older and middle-aged adults may benefit from assistance provided by an exoskeleton, and as gait changes have been observed as people age [4], it is important to consider the gait patterns and performance of middle-aged and older adults when designing exoskeletons.

In particular, gait-related data on middle-aged adults is lacking [7]. Characterizing the effects of added mass on the gait of middle-aged adults is important, as changes in gait performance, such as slower speed, shorter step length, narrower stride width and variability in step length have been correlated with an increased risk of falling and injury [8]. Quantifying the changes in gait performance of middle-aged adults in response to added mass could help exoskeleton designers account for these changes and design safer and more efficient exoskeletons for this population. Despite the recent advances in exoskeleton technology, there is still a gap in knowledge about how humans might benefit from the use of an exoskeleton [9]. To accurately quantify and improve the performance of an exoskeleton, we must first be able to quantify human performance [9]. This is especially true in the case of “assist as needed” control algorithms [10,11]. In the case of exoskeletons that employ “assist as needed” control strategies, it is expected that the user can walk unhindered when the assistance is not active. However, the additional mass will affect the gait of the user [12,13,14,15]. To improve the performance of such exoskeletons, it is important to quantify the effects of added mass across the whole gait cycle. All exoskeletons that have a “zero-impedance” or “patient-in-charge” setting will also benefit from better characterization of the user’s starting point [10,16]. Specifically, SPM can be used to quantify the gait changes across the whole gait cycle. The regions of no change can also help inform exoskeleton designers about the regions in the gait cycle where active control is not required. Previous studies have noted the importance of quantifying the human response to mechanical assistance, as this could in part reduce metabolic cost and make the exoskeleton design more efficient [9,17,18,19].

Prior studies have also focused mainly on a cohort of healthy younger adults to characterize the effects of added mass on gait [14,20,21]. To our knowledge, there is no study that quantifies the effects of added mass on middle-aged adults. Nevertheless, aging is related to changes in the musculoskeletal system as well as changes in the central and peripheral nervous systems [7]. These changes in musculoskeletal capabilities and proprioceptive feedback could affect an older individual’s ability to tolerate and adapt to the added mass indicative of an exoskeleton. This study will focus on the gait-related adaptive behaviors of middle-aged adults when mass is added around the waist and the thigh. As such, this study’s design will help bolster the basic science behind the design of hip exoskeletons and provide insight into how middle-aged adults respond to walking with added mass.

Most prior studies have used discrete point analysis, often focusing on the gait peaks or gait events to identify the effects of added mass on walking [12,13,20]. This introduces a selection bias, and it does not allow for a fully representative analysis of the effects of added mass across the whole gait cycle. Statistical parametric mapping (SPM) is a technique that has previously been used to analyze biomechanical metrics across the whole gait cycle [22,23]. A prior study on the effects of added mass on gait kinetics and kinematics has indicated that mass of up to 16 kg is well tolerated on the pelvis and thigh [12]. However, this conclusion is based on the analysis of gait at discrete points of reference. Further investigation is required to quantify the effects of added mass on gait characteristics across the whole gait cycle.

The purpose of this study is to characterize the effects of pelvis- and thigh-added mass on gait parameters across the whole gait cycle. This study aims to provide insights into different potential exoskeleton mass distribution patterns, and the responses of both young and middle-aged adults to these mass distribution patterns. We hypothesize that loading at the pelvis and thigh will have different characteristic impacts on gait parameters, with pelvis loading having a greater impact on stance and thigh loading having a greater impact on swing. Based on prior literature, we also expected the young and middle-aged adults to have different responses to the added mass during walking [7]. Therefore, we examined gait parameter differences between young and middle-aged adults in response to generalized added mass. For this, we grouped all added mass conditions, which reflects the reality that exoskeletons have a variety of possible configurations. The results of this study will provide information on how young and middle-aged adults respond to the added mass of wearing an exoskeleton and how sensor-guided exoskeleton control can be designed to provide gait assistance to exoskeleton users.

## 2. Materials and Methods

### 2.1. Participants

The study recruited 32 participants without any lower limb injuries, and data from 4 participants were removed due to technical difficulties. Therefore, 14 healthy young participants (6 females, aged 22 ± 4 years, 71.0 ± 10.5 kg body mass, 1.72 ± 0.07 m height and 23.98 ± 3.52 kg/m^2^ BMI) and 14 healthy middle-aged participants (7 females, aged 48 ± 9 years, 76.5 ± 13.0 kg body mass, 1.72 ± 0.08 m height and 26.12 ± 4.16 kg/m^2^ BMI) were included in the analysis. The research protocol was approved by the University of Dayton’s Institutional Review Board, and all participants provided written informed consent.

### 2.2. Data Collection and Processing

The participants were provided the same style of running shoes. The main body segments considered for this study were the foot, shank, thigh, and pelvis segments. Standard lower extremity bony landmarks (full list in Figure 1) were palpated and marked with passive retroreflective markers for the collection of motion capture data [24]. Additional markers were applied only for tracking purposes, such that a minimum of at least 4 markers would be used to track each considered body segment. A consistent approach was followed for placing the markers on all subjects, and markers were placed such that they did not interfere with the added masses or the added mass wraps. A total of 33 markers were used, and the full marker set is shown and described in Figure 1. A static trial was recorded prior to each loading condition in which the participant stood with feet flat on the ground, pelvis level and knees straight. Surface electromyography (EMG) sensors were placed on medial gastrocnemius, soleus, tibialis anterior, vastus lateralis, vastus medialis, biceps femoris and semitendinosus muscles on the right leg, following SENIAM recommendations [25].

A two-minute overground walking trial was used to determine a moderate self-selected walking speed. Walking speed was 1.24 ± 0.16 m/s for the younger group and 1.19 ± 0.13 m/s for the middle-age group. Prior to data collection, participants walked for 3 min on the treadmill to acclimatize. The participants walked at their selected speed on the instrumented treadmill for 1 min for each of the 9 different loaded conditions. Additional masses were added bilaterally to the thigh and pelvis in combinations defined in Table 1. Conditions with added mass placed only on the pelvis or thigh were not considered for this study, as the mass distribution is generally not isolated on one segment in a hip exoskeleton [26]. Cylindrically shaped, high-density tungsten bucking bars (1.0″ diameter, 3.6″ length) were used to add the mass symmetrically to the subjects (Midwest Tungsten Services, Willowbrook, IL, USA). The added mass was placed in a padded, cylindrical holder. Coban wraps were used to wrap the added mass around the thigh segment, and a waist belt was used to place the added mass on the pelvis. The added mass conditions were randomized for each subject.

The marker trajectories were recorded at 150 Hz with a 10-camera Vicon motion capture system (Oxford Metrics, Oxford, UK). Ground reaction force data were collected at 1500 Hz using two force plates in an instrumented split-belt treadmill (Bertec Corp., Columbus, OH, USA) and muscle activity data were collected at 1500 Hz, using the Delsys Trigno wireless EMG system (Delsys Inc., Boston, MA, USA). Any gait cycles without a clean right foot strike on the right force plate and a clean left foot strike on the left force plate were not considered for the analysis. To reduce the artifacts in joint moments and joint reaction forces that might be caused by differences in the frequency content of the kinematic and kinetic data, both the marker and force plate data were smoothed using a lowpass filter with a cutoff frequency of 15 Hz [27,28,29].

### 2.3. Data Analysis

#### 2.3.1. Joint Kinematics and Kinetics

Visual 3D software (C-motion Inc., Germantown, MD, USA) and MATLAB (The MathWorks^®^, Natick, MA, USA) were used for biomechanical modeling and analysis. Lower body models were generated with segment coordinate systems based on the bony landmarks represented by the applied motion capture markers [24]. The pelvis segment was modeled using the Visual 3D composite approach, which is based on the markers at the posterior superior iliac spine and anterior superior iliac spine bony landmarks [30].

The functional hip joint centers of the models were determined from a range of motion trial, using principles of the Gillette algorithm [31]. Knee joint centers were defined as the midpoint between the lateral and medial femoral epicondyle markers. Ankle joint centers were defined at the midpoint between the fibula apex of the lateral malleolus and the tibia apex of the medial malleolus.

In the segment coordinate system, the sagittal axis is defined as X, the frontal axis is defined as Y, and the transverse axis is defined as Z [30,32]. The transverse axis for the thigh is the vector from the knee joint center to the hip joint center. The sagittal axis for the thigh is the vector between the lateral and medial femur epicondyle markers. The transverse axis for the shank is the vector from the ankle joint center to the knee joint center. The sagittal axis for the shank is the vector between the fibula apex of the lateral malleolus and the tibia apex of the medial malleolus. The segment coordinate systems follow the right-hand rule, such that the transverse axis is positive in the superior direction, the sagittal axis is positive towards the right side, and the frontal axis is positive in the anterior direction [30]. In quiet stance (static pose), the coordinate system primary axes are generally aligned, and joint angle changes during walking are calculated using the Cardan sequence order sagittal-frontal-transverse [32].

The sagittal plane joint angles, joint internal moments and joint reaction forces for the hip, knee and ankle joints were calculated within Visual 3D (v2021, C-Motion Inc., Germantown, MD, USA). For the baseline model mass distribution, the Visual 3D default percentages of total body mass were used for each segment [33]. For each loading condition, the appropriate additional mass was added to the baseline segment masses of the Visual 3D model. The joint moments and the joint reaction forces were normalized by the average mass of the respective subjects, calculated from a baseline static trial with no added mass. The default settings within Visual 3D were used to calculate the moment of inertia [34] and the center of mass [35] of all segments. The joint angles, joint moments and joint reaction forces for each gait cycle were time-normalized to 101 points (for a total of 9828 gait cycles per metric).

#### 2.3.2. Muscle Activity

The raw EMG data were bandpass filtered between 20 Hz and 500 Hz [36]. The signal was then full wave rectified, and a linear envelope was generated by low pass filtering at 12 Hz, using a 4th order Butterworth filter. For each subject, the linear envelopes were normalized by the average of the peak values from the 60 s trial with the highest amount of added mass on both segments (+10.8 lb on the pelvis and +5.4 lb on each thigh). The EMG data from each gait cycle was then time-normalized to 101 points.

#### 2.3.3. Statistical Parametric Mapping

Statistical parametric mapping (SPM) was used to compare the gait parameters (joint angles, joint moments, joint reaction forces and muscle activity) of middle-aged adults to young adults. All SPM analyses were conducted using the open-source MATLAB software package from spm1D 0.4 [23]. A 1-way repeated measures ANOVA, as described by Pataky, was used to evaluate the effects of age (young or middle-aged) on the gait parameters separately [23]. Two 2-way repeated measures ANOVAs were used to evaluate the effects of pelvis load (low, medium, or high) and thigh load (low, medium, or high) on the gait parameters separately for both the younger cohort and the middle-aged cohort [23]. Post hoc paired *t*-tests were used to compare pairs of conditions. The significance level for all statistical tests was set to 0.05. In comparing the means, regions where the calculated magnitude of the *t*-statistic was greater than the t critical value were marked as statistically significant. The *t*-statistics are qualitatively identical to effect size and can be used as an indicator of practical significance. The SPM results have only been verified for balanced datasets. To make the dataset balanced, gait cycles were removed from the analysis, in a pair-wise manner in the case of crossovers on the force plates, poor marker tracking or bad EMG signals. An EMG signal was determined as unusable or bad if the data showed a constant static noise, with a signal peak-to-baseline ratio lower than 2:1.

## 3. Results

The results of SPM are often interpreted using the different phases of the gait cycle as a reference [22,23]. Figure 2 shows a typical gait cycle and the phases that will be referred to in the study. Figure 3 and Figure 4 represent the SPM results used to evaluate the effects of the pelvis and thigh mass on the gait of both the middle-aged cohort and the younger cohort. These summary figures allow for the overall signature of pelvis loading and thigh loading on gait metrics to be visualized. However, several regions of significant difference identified by the SPM analysis occur where the metric magnitudes are close to zero. We present here the full results of the SPM and have not removed any results based on additional thresholds or other criteria. The results are best interpreted using Figure 5, Figure 6, Figure 7, Figure 8, Figure 9 and Figure 10, which includes the profiles of each gait metric at the high mass level compared to the low mass level. The regions of significant differences from the SPM results are noted in each gait metric profile plot with background shading (Figure 5, Figure 6, Figure 7, Figure 8, Figure 9 and Figure 10). In combination with the gait metric profiles and shaded regions denoting significant differences, the SPM {t} value plots directly below each gait metric plot (Figure 5, Figure 6, Figure 7, Figure 8, Figure 9 and Figure 10) can be interpreted as a visual representation of the effect size. While SPM is designed to identify regions of significant differences, the points of greatest difference can be identified at the points in the gait cycle, where the SPM {t} plot reaches a local maximum or minimum.

### 3.1. Overall Response of Young Adults to Pelvis Mass Increases

The SPM results shown in Figure 3 and Figure 4 represent the changes due to increasing mass. The significance mapping figure for the young adults (Figure 3) shows that even for a lower amount of pelvis mass (pelvis: low-medium frame) there are changes in joint reaction force at every lower limb joint for the majority of stance, and changes in knee and hip angles are present in early stance and late swing. Further comparison of changes at higher levels of pelvis mass (medium-high and low-high) shows that these basic changes in reaction force and knee and hip angles expand across larger sections of the gait cycle. For joint moment metrics, significant regions of change are more apparent for comparisons of large mass change (pelvis: low-high frame), which shows increased hip and knee moments in mid-stance (Figure 3, Figure 7 and Figure 9) and increased ankle moments in mid-stance through toe-off (Figure 3 and Figure 5). While SPM results show significant changes in joint reaction force during swing, all magnitudes are near zero. Joint moment changes during swing also require a more nuanced interpretation (see discussion). Muscle activity changes with pelvis loading can be summarized as the hamstrings group (biceps femoris and semitendinosus) increases in early stance (Figure 9), vasti group (vastus lateralis and medialis) increases in mid-stance through swing (Figure 7) and soleus increases in late stance (Figure 5).

### 3.2. Overall Response of Middle-Aged Adults to Pelvis Mass Increases

For middle-aged adults, the significance mapping figure (Figure 4) shows that even for a lower amount of pelvis mass (pelvis: low-medium frame), there are changes in the joint reaction force at every lower limb joint for the majority of stance, which is consistent with the younger group. Even for lower pelvis mass, ankle moment was significantly increased in the region of late stance. Further comparison of changes at higher levels of pelvis mass (medium-high and low-high frames) shows that these basic changes in reaction force expand across larger sections of the gait cycle. Only very short regions of change are significant for ankle angle (late swing) and knee angle (heel strike transition), while hip angle does increase significantly in mid-stance. For joint moment metrics, significant regions of change become more apparent with higher levels of pelvis mass. Particularly, the ankle moment increase was present across a longer mid-stance to toe-off duration (Figure 6). Regions of increased knee moment were observed in mid-stance and at toe-off for the middle-aged group (Figure 8). The middle-aged group had short regions of change in the hip moment during swing (Figure 10). Muscle activity changes with pelvis loading can be summarized as the vasti group increases in early stance (Figure 8) and some tibialis anterior increases in early stance and mid-swing (Figure 6).

### 3.3. Overall Response of Young Adults to Thigh Mass Increases

For young adults, the significance mapping figure (Figure 3) shows that even for a lower amount of mass (thigh: low-medium frame), there are changes in ankle and knee reaction forces for the majority of stance, and hip reaction force increases are also present. While SPM results show significant changes in the joint reaction forces during swing, all magnitudes are near zero. Joint angle changes were limited but did include decreased ankle and knee angles in early stance. At higher levels of thigh mass (medium-high and low-high), changes during toe-off (60–70% of the gait cycle) can be characterized as a more plantarflexed ankle (toe pointed), a more flexed knee (bent) and a less extended hip (more under the pelvis, increased hip angle) (Figure 5, Figure 7 and Figure 9). This corresponds with regions of significantly increased hip and knee moments. Changes during swing (70–90%) can be characterized as a more dorsiflexed ankle, a more extended knee (straight), and a less flexed hip (less forward, decreased angle). This corresponds with regions of significantly decreased hip and knee moments. Muscle activity changes with added thigh mass can be summarized as hamstrings group (Figure 9), gastrocnemius and soleus increase during stance, and tibialis anterior increases during mid-swing (Figure 5).

### 3.4. Overall Response of Middle-Aged Adults to Thigh Mass Increases

The significance mapping figure (Figure 4) shows that even for a lower amount of mass (thigh: low-medium) in the middle-aged group, there are changes in ankle and knee reaction forces for the majority of stance, and hip reaction force increases are also present. Further comparison of changes at higher levels of thigh mass shows consistently increased joint reaction forces throughout stance for all joints (Figure 6, Figure 8 and Figure 10). In late stance, there are increases in ankle and knee angle (Figure 6 and Figure 8). During toe-off and swing, the middle-aged group shows the same characteristics of joint angle changes as described for the younger group. Ankle moment increases during stance for a small region of the gait cycle (Figure 6). The hip moment increase shown for the middle-aged group is very short and occurs only during early stance (Figure 10). Muscle activity changes can be summarized as decreases in the vasti group during early stance (Figure 8), increases in gastrocnemius and soleus during mid-stance, and tibialis anterior increases during mid-swing (Figure 6).

### 3.5. Profile Differences across the Three Considered Joints

Figure 5, Figure 6, Figure 7, Figure 8, Figure 9 and Figure 10 represent the SPM results used to evaluate the effects specific to pelvis and thigh mass on the gait of both the younger cohort and middle-aged cohort. Recall that a full-factorial approach with three levels was used for the mass combinations tested. These figures focus on the low-high comparisons. For the pelvis, this compares conditions 1, 4 and 7 (pelvis low mass) against conditions 3, 6 and 9 (pelvis high mass) (Table 1). For the thigh, this compares conditions 1, 2 and 3 (thigh low mass) against conditions 7, 8 and 9 (thigh high mass) (Table 1).

### 3.6. Age Related Differences in Gait Metrics

Figure 11, Figure 12 and Figure 13 show the differences in gait kinematics, kinetics and muscle activity between the two age groups. Data included in these analyses and plots include the responses to all added mass conditions (Table 1) and represent generalized responses to the added mass of an exoskeleton. As the mean walking speeds for the groups differed by only 0.05 m/s (young cohort: 1.24 m/s; middle-aged cohort: 1.19 m/s) differences between the two groups should represent differences in response rather than walking speed. All the considered gait metrics were significantly different between the two age groups under loaded walking. The ankle angle, knee angle and hip angle were significantly different in 86%, 99% and 88% of the gait cycle, respectively. The ankle reaction forces, knee reaction forces and hip reaction forces were significantly different in 94%, 92% and 95% of the gait cycle, respectively. The ankle moment, knee moment and hip moment were significantly different in 89%, 91% and 73% of the gait cycle, respectively. The gastrocnemius had significant activation differences in 51% of the gait cycle, and all other muscles had activation differences in at least 90% of the gait cycle.

## 4. Discussion

Responses are discussed specific to the location of the added mass and phase of the gait cycle in the following way: 4.3 pelvis loading stance impact, 4.4 pelvis loading swing impact, 4.5 thigh loading stance impact, 4.6 thigh loading toe-off impact, and 4.7 thigh loading swing impact. Within each section, the discussion will examine the young adult responses and then the middle-aged adult responses, noting qualitative similarities and differences. Relevant statistical differences between the age groups’ overall responses to loading will be discussed. Finally, each paragraph or section will conclude with the implication of the response on the design of exoskeleton systems or control approaches.

### 4.1. Characterizing Response Signatures for Added Mass at Pelvis and Thigh

The focus of this study is to understand the impact of exoskeleton weight for exoskeletons that cross the hip joint and attach primarily at the pelvis and thigh. Depending on exoskeleton functionality and design choices, a variety of weight distributions between the pelvis and thigh might be selected [26]. The fact that the location of an added mass (e.g., pelvis vs. thigh) will have a specific impact on gait parameters is well established [12,14]. Our results further this knowledge with the application of the SPM analysis approach, which allows the impact of mass on the entire gait cycle to be considered in contrast with prior work focused on peaks or specific instances in the cycle [12,13,20]. This study supports the idea that the greatest effects of added mass (SPM {t} values) generally occur at the local minima or maxima in the metric profile. However, researchers are aware that differences extend beyond those particular instances. In the case of exoskeletons, control strategies should seek to smoothly modify applied assistance leading up to and following those peak instances. Consideration of the SPM {t} value plots may improve the capability of exoskeleton designers to determine appropriate ranges in the gait cycle to begin and end applied assistance, such that the desired profile can be achieved. SPM results allow the signature of the impact to be fully visualized both across the gait cycle and for many parameters in a single figure. For both young and middle-aged adults, it was observed that the response to the pelvis loading had different characteristic features compared to thigh loading. This allows exoskeleton researchers to understand how the magnitude and allocation of mass in their device are likely to impact the user throughout the complete gait cycle.

### 4.2. Comparing to Studies with Other Mass Magnitudes or Increments

A challenge in this area of study is that research groups have used a variety of different magnitudes of added mass and different increments of added mass [12,13,21,37]. The results of the SPM mapping suggest that the signatures of the response to added mass will be relatively similar when comparing responses at a lower magnitude (low-medium) and a higher magnitude (medium-high) or when comparing different increments of change in mass (medium-high vs. low-high). While thresholds clearly exist for the lowest mass that can produce a perceptible change in a metric, within the range of masses used in real exoskeletons, the human response likely keeps the same characteristics regardless of the exact mass amount. Specifically, as added mass is increased, SPM shows that regions of significant changes seen even at low mass levels are generally maintained but increase in duration over the gait cycle (Figure 3 and Figure 4).

### 4.3. Response to Pelvis Loading during Stance

#### 4.3.1. Young Group

As expected, increased pelvis loading increases joint reaction forces throughout stance for both young and middle-aged groups (Figure 3 and Figure 4). The younger group increased their hip and knee angles from heel strike through mid-stance and increased the moments of these joints across a similar region of stance. This increased hip flexion in early-to-mid stance likely represents a strategy for further forward placement of the striking limb to better absorb and redirect the increased mass/momentum at the pelvis. The maximum hip moment requirement occurs in early stance, and the young group generated significantly higher hip moment throughout the majority of this peak (Figure 13). Some observed increases in early stance hamstring activity (but not vasti activity) may be related to these hip moment changes (Figure 3 and Figure 9). In knee angle kinematic profiles, the smaller flexion peak observed during early stance is frequently interpreted as a weight acceptance response [38]. The increased flexion and concurrent increase in knee moment during early stance observed in our younger cohort appear to be a reasonable response to added mass.

#### 4.3.2. Middle-Aged Group

The middle-aged group showed a markedly different response, with hip kinematic changes greatly reduced in duration (early stance only) (Figure 3 and Figure 4). Age group comparison for the hip shows that the young group maintains greater hip flexion and extension peaks when loaded compared to middle-aged responses (Figure 13). The middle-aged group also did not demonstrate the increased knee flexion shown by the younger group during weight acceptance (Figure 7 and Figure 8). Age group comparison for the knee angle highlights that the middle-aged group maintains a more extended knee angle profile than the younger group throughout stance (Figure 12). Prior work describes this weight acceptance (or energy absorption) knee behavior as being controlled by the eccentric contraction of knee extensors [38,39]. This is followed by concentric knee extensor contraction, creating an extension moment to straighten the knee [38]. Increased muscle activation in both vasti muscles from roughly 5–25% of the gait cycle supports this interpretation that additional load requires additional knee extensor recruitment during weight acceptance, particularly with the middle-aged group not allowing increased knee flexion (Figure 4 and Figure 8). Overall, this suggests that the middle-aged group is more resistant to making kinematic adaptations than the younger group and makes no kinematic modifications to prepare for heel strike or accept additional pelvis mass.

#### 4.3.3. Weight Acceptance Behaviors

The small region of hip angle increase (12–30% of the gait cycle) corresponded with the maximum hip moment requirement, but unlike the young group, the middle-aged cohort did not significantly change their hip moment (Figure 10). While knee joint kinematics were unchanged, for the middle-aged cohort, the knee joint moment (flexion) was increased throughout the region of early stance knee moment peak (12–32% of the gait cycle) (Figure 8). In the age group comparison, for generalized added mass, knee joint angles and knee moments were significantly lower at this point in the gait cycle for the middle-aged cohort compared to the younger cohort (Figure 12). Keeping a straighter limb during weight acceptance appears to allow for less knee moment production, but at the cost of generating a greater reaction force loading rate at the knee and hip (Figure 12 and Figure 13). The middle-aged group may be avoiding kinematic adaptations to reduce their knee moment requirement, but in a way that can cause long-term injury or osteoarthritis. Prior literature supports the idea that older adults demonstrate reduced hip range of motion and moment production capabilities [40,41]. These kinds of changes in gait behaviors and strategies with increasing age highlight how beneficial exoskeleton use could be for supporting decreased moment generation capacity and encouraging an increased range of motion in older adults. However, if not implemented with an understanding of the middle-aged differences, the older adult user may struggle to adopt the desired kinematic changes that could improve their gait pattern and their ability to mitigate the added load of an exoskeleton. Particularly, if the exoskeleton sensor packages and control schemes are unable to monitor and address altered joint reaction forces, it could be a concern for long-term users.

#### 4.3.4. Ankle Joint Responses

Neither group modifies their ankle angle during stance in response to pelvis loading, but both groups produce increased ankle moment from mid-stance to toe-off (35–65% of the gait cycle), which is the peak area of the profile (Figure 5 and Figure 6). While adaptations occur for both groups, the age group comparison highlights that peak ankle moment and peak plantarflexion are lower for the middle-aged group compared to the younger group (Figure 11). The late-stance phase represents ankle power generation and is reported to contribute to roughly half of the positive work in a stride [42,43,44]. If pelvis loading increases ankle moment demand, but middle-aged adults have lower production capability, this should be a consideration in the design of exoskeletons that target the hip and knee. Exoskeletons with mass at the pelvis are likely to generate similar responses in the ankle joint. However, hip and knee exoskeletons would likely have no capability to provide assistive torques at the ankle and decrease the burden on the user to generate increased ankle moments. While further study is needed, it is likely that with exoskeleton sensors and a study of control approaches, the behaviors at the hip and knee could be modified to reduce the impact on the ankle joint.

### 4.4. Response to Pelvis Loading during Swing

The younger group increased their hip flexion in early swing and late swing and increased their knee flexion in late swing in response to pelvis loading (Figure 3, Figure 7 and Figure 9). These behaviors may represent preparatory adaptations to better position the foot relative to the pelvis and to better accept the increased pelvis load. Conversely, the middle-aged group makes no hip angle changes (Figure 4 and Figure 10). They do increase knee flexion briefly just prior to and after heel strike, but this is only approximately 5% of their total gait cycle in comparison to the younger group, who maintain this significant increase over closer to 50% of their total gait cycle (Figure 7 and Figure 8).

During mid-swing (at approximately 80% of the gait cycle), both the young group and the middle-aged group significantly increase their tibialis anterior activation with added pelvis mass, which would serve to raise the toe and maintain swing leg clearance [45]. While toe-clearance measurements were beyond the scope of this work, this is a reasonable co-behavior to the increased knee flexion observed during weight acceptance (mid-stance on the opposite leg). The young group clearly increased knee flexion, which would lower the pelvis and increase the requirement for toe-raise in the swing leg. While the middle-aged group did not demonstrate increased knee flexion in their response, it could be an issue of perception or require further analysis of the responses in a world coordinate frame. This outcome is also observed for thigh loading and is further discussed there.

### 4.5. Response to Thigh Loading during Stance

#### 4.5.1. Response at Low Thigh Loads

In contrast to pelvis loading, the observed responses to thigh loading appear as an exception to the proposed idea that the impact of loading on metrics is relatively similar despite the net magnitude or increment considered. The responses to loading a particular location generally have similar characteristics for low-medium, medium-high and low-high. However, the thigh low-medium responses do entirely lack some regions of significant differences observed in the other comparisons (medium-high and low-high) (Figure 3 and Figure 4). This is particularly true for the younger group and the kinematic results (Figure 3). This suggests that low-magnitude mass on the thigh can be easily managed by young and middle-aged adults and does not necessitate or favor discernable kinematic adaptations. Prior work has described the idea that thresholds exist below, in which added mass does not generate significant changes [12,14]. This may represent a benefit for exoskeletons that focus on the hip joint but require some attachment at the thigh. A low thigh mass consistent with an exoskeleton attachment appears to result in a few regions of significant change. Given this, the rest of the discussion will focus on the response patterns observed in the thigh: medium-high and low-high comparisons.

#### 4.5.2. Weight Acceptance Behaviors

Both groups showed increased ankle and knee angles prior to heel strike, and these changes in response to thigh loading were maintained in early stance (Figure 3 and Figure 4). Just before and after heel strike, the middle-aged group significantly increased muscle activity in the gastrocnemius, tibialis anterior and vasti groups (Figure 4, Figure 6 and Figure 8). This was not observed in the younger group (Figure 3, Figure 5 and Figure 7) and appears to represent a general strategy of stiffening the limb to accommodate the additional thigh mass. As muscle forces are the main determinant of joint loading [46], the observed stiffening may have a negative impact on joint loading under added mass or exoskeleton use conditions. Both groups had significantly increased joint reaction forces from initial loading and throughout stance, but with hip, joint reaction forces significantly impacted over slightly reduced regions compared to the other joints (Figure 9 and Figure 10). While loading rates are not presented explicitly, the joint reaction force profiles (Figure 5, Figure 6, Figure 7, Figure 8, Figure 9 and Figure 10) capture that increased thigh mass contributed to steeper profile curves during initial loading for all joint reaction forces. The age group comparison figures highlight that joint reaction force profiles for the middle-age group were also initially steeper than for the young group (Figure 11, Figure 12 and Figure 13). This could be a concern for all exoskeleton users, but particularly for middle-aged and older users, in light of a possible stiffening strategy that may also alter joint loading.

#### 4.5.3. Mid-Stance Kinematics

In mid-stance, we note that the younger group increased their ankle, knee and hip angles at approximately 40% of the gait cycle, with the knee joint appearing most sensitive to the added thigh mass (Figure 5, Figure 7 and Figure 9). This point in the gait cycle represents the thigh being almost directly under the pelvis and the knee angle moving to its most extended position. The middle-aged group had similar timing and duration for ankle and knee increases (Figure 6 and Figure 8), but did not alter their hip angles (Figure 10) in response to the added thigh mass. Overall, this suggests that accepting a slightly less extended leg at mid-stance is a natural response regardless of age to the increased thigh mass on both the stance and swing legs. This appears to coincide with increased efforts to clear the swing leg, which are discussed under pelvis loading and are also addressed under the thigh loading swing response.

#### 4.5.4. Ankle Joint Responses

For both groups, the ankle joint moment was most sensitive to thigh loading, with increases across stance for longer durations versus pelvis loading (Figure 5 and Figure 6). Consistent with this was the increased activity in the gastrocnemius and soleus for both groups (Figure 5 and Figure 6), which would contribute to ankle plantarflexion [47]. It is useful to note that the regions of increased ankle moment and plantar flexor activity identified by the SPM analysis did not correspond to the metric peaks and, therefore, might not be observed in a standard analysis of discrete points in the cycle. As noted in the pelvis loading section, hip joint exoskeletons with mass concentrated at the pelvis and thigh are likely unable to support increased ankle moment requirements directly but may be able to influence hip, knee, or overall gait behaviors to lessen the potential impact on the ankle.

#### 4.5.5. Joint Moments and Muscle Activity

For the middle-aged group, hip moment was significantly increased for the majority of the first 20% of the gait cycle with added thigh mass (Figure 10). For the young group, hip moment was significantly increased across most of the stance phase (Figure 9). For both groups, a deviation in the hip moment profile at roughly 10% of the gait cycle is apparent. Typically, the initial hip extension moment peak is the greatest in magnitude, and the secondary peak at toe-off of the trailing limb is identifiable at 10% of the gait cycle but lower in magnitude. In this part of the gait cycle, the hip joint moment is recognized as a concentric hip extensor contraction, which generates energy, manages loading response and controls the body’s forward acceleration [38,48]. In the case of thigh loading, when trailing limb toe-off occurs, the stance leg hip moment is recruited both to stabilize the added mass in the stance leg and pull the added mass in the swing leg forward. In the young group, a large profile change in semitendinosus activity during this time and some biceps femoris increases support this idea (Figure 9). However, in the middle-aged group, hamstring muscle group response is reduced at this time (Figure 10), which suggests that they employ a different strategy, possibly due to difficulty in rapidly shifting from vasti group activation (for stiffening the limb) to hamstring group activation (for hip extensor moment generation). Hip exoskeletons should be capable of providing an external hip extension moment during this toe-off transition; however, exoskeletons lacking sensors at the foot will likely need to employ torque or muscle activity sensors to correctly time this added support.

### 4.6. Response to Thigh Loading during Toe-Off

#### 4.6.1. Impact of Temporal Shifts

In contrast to pelvis loading, thigh loading resulted in gait metric modifications that were specific to the toe-off region of the gait cycle (roughly 60–70% of the gait cycle) (Figure 3 and Figure 4). For the thigh: low-high comparison, both groups demonstrated a more plantarflexed ankle (toe pointed, decreased angle) (Figure 5 and Figure 6), a more flexed knee (bent, increased angle) (Figure 7 and Figure 8), and a less extended hip (more under the pelvis, increased angle) (Figure 9 and Figure 10). While the younger group showed some muscle activation changes during toe-off, the magnitudes of activation for those muscles were very low, so it is unlikely for them to have much impact (Figure 5 and Figure 9). Inspection of the profile figures suggests that rather than a change in the gait kinematics, this appears visually as more of a temporal shift of the toe-off event. Particularly, thigh loading appears to favor a smaller stance-swing ratio, meaning a longer swing time with a toe-off at a slightly earlier percent of the gait cycle. The interpretation of temporal stance-swing adaptations is a challenge for all biomechanics studies that use normalized gait cycle plots. For the SPM approach, these changes can potentially be misinterpreted, but in combination with the kinematic profiles, they can accurately identify where the timing of a gait metric is altered. Further investigation is required to determine if SPM analysis should normalize stance and swing independently to better characterize gait adaptations.

#### 4.6.2. Joint Moments

For example, SPM reports both increased and decreased hip and knee moments during and just after the toe-off region (Figure 3 and Figure 4), but inspection of the figures suggests this is the result of a temporal shift rather than an adoption of higher or lower moment magnitudes due to loading (Figure 7, Figure 8, Figure 9 and Figure 10). The pattern of hip and knee moments during toe-off is relatively complex, and the ability of an exoskeleton to sense the appropriate timing for effectively applying these moments is critical to success and usability. The age group comparison shows that the middle-aged group generated significantly more hip flexion moment (more negative values) during toe-off (Figure 13), which appears to compensate for the reduced ankle plantarflexion moment just prior to swing and the reduced ankle joint plantarflexion (Figure 11). This suggests that exoskeletons might compensate for the lack of ankle actuation with assistive hip moment application.

### 4.7. Response to Thigh Loading during Swing

#### Kinematics and Clearance

While SPM analysis reported statistical changes in the joint reaction force during swing, these likely had minimal impact due to the force magnitudes being close to zero overall (Figure 5, Figure 6, Figure 7, Figure 8, Figure 9 and Figure 10). For kinematics, the characteristic swing response to thigh loading for all groups and comparisons is a more dorsiflexed ankle (increased angle), and a more extended knee (straight, decreased angle) (Figure 3 and Figure 4). Significant changes in the ankle angle during toe-off in both groups suggest a slightly earlier toe-off, where for a given point in time relative to the gait cycle, the toe-off behavior is further advanced when increased thigh loading is present. The increased ankle angle during swing potentially represents both a shifting of the kinematic response and some increased demand for toe-clearance due to a straighter knee joint angle. The middle-aged group particularly appears to: (1) shift their behavior temporally and (2) achieve reduced peak knee flexion values in response to thigh loading (Figure 8), compared to the younger group (Figure 12). Consistent with the increased toe-clearance requirement, both groups increased their tibialis anterior activation throughout mid-swing (Figure 5 and Figure 6). The potential impact of exoskeletons that add mass to the pelvis and thigh generates two responses that will impact toe-clearance. First, pelvis loading may cause increased knee flexion as a weight acceptance response, lowering the pelvis overall. Second, thigh loading may result in decreased knee flexion in the swing leg and create a longer effective leg length during swing. Given this, the major ankle angle adaptations demonstrated by middle-aged adults may be more related to toe-clearance requirements than age-induced limitations, and thus appear important to consider in future work. Even pathological middle-aged adults have been shown to be adept at modifying their gait to maintain toe-clearance, but long-term use of an assistive device should relax any unneeded toe-clearance behaviors [49].

A potential benefit of increasing the swing percent of the gait cycle is that the thigh segment can be moved forward (or hip angle increased) at a slower rate. Both groups appear to adopt a lower rate of change in hip angle during swing (Figure 9 and Figure 10). The middle-aged group also demonstrates reduced peak hip flexion angles in later swing, which could be interpreted as either a result or a facilitation of the reduced rate. Finally, just prior to heel strike, both groups significantly increased their hip extension moments in response to thigh loading, which appears to be the additional requirement to decelerate the additional mass on the forward swinging leg [38].

### 4.8. Limitations

The thigh mass was distributed asymmetrically (Figure 1). More mass was placed laterally, and this could partly influence some of the thigh mass-related changes that were noted in the study. More mass was distributed on the lateral side of the thigh to mimic the build of most exoskeletons [15,26]. This study also does not consider the muscle activity of any hip flexors. The motion of the three joints considered, in the sagittal plane, could have been more completely defined with the inclusion of a hip flexor, such as the rectus femoris. However, this was not conducted in this study, as the placement of the added mass interfered with the collection of clean EMG signals from the rectus femoris. SPM has also not been verified for unbalanced three-way repeated measures ANOVA. To make our dataset balanced, we had to remove some viable data in a pairwise manner. The removed gait cycles were randomly selected, and even the factor with the least number of gait cycles had at least 4576 gait cycles of data analyzed. SPM does not consider the possible differences in the timing of gait events, and this could partly be responsible for some of the changes reported in this study. Nevertheless, the comparisons between middle-aged adults and younger adults are still valid, as the differences in mean gait speed were only approximately 0.05 m/s. We acknowledge the possibility that our study could be underpowered at certain regions of significance, but we believe our sample size is sufficient following the benchmarks from Luciano et al. [50]. We also note that our study is appropriate for comparison to similar biomechanical studies of added mass, with sample sizes varying from 5 to 12 subjects [12,13,20,21].

## 5. Conclusions

Lower limb exoskeletons might differ in their anticipated functional use, ranging from full movement support to negating the impact of heavy loads or providing targeted assistance. However, almost all functions provide a torque on the limb that is transferred from a motor to a joint in the body [2,3]. As the motors, power sources and attachments themselves can be heavy; an overarching design goal is that the exoskeleton be capable of offsetting the impact of its own added mass to the greatest extent possible [15,26]. Exoskeletons with “assist-as-needed” or “zero impedance” control strategies may benefit from the SPM analysis approach, which highlights the regions of the gait cycle over which particular joint assistance is likely to be required. Control strategies, in combination with sensor feedback, will determine if these applied torques are used to guide/enforce kinematic patterns, assist or resist joint moments produced by the user, predict and respond to events, such as heel strikes, and avoid dangerous events, such as falls.

In this study, we sought to characterize the impacts of added mass specific to loading at the pelvis location and at the thigh location. The same protocol was examined separately for a young adult and a middle-aged adult cohort. In each cohort, we found characteristic gait metric changes in response to pelvis-added mass and to thigh-added mass. For both added mass locations (pelvis and thigh), the regions of significant differences are generally maintained, but increased in duration with an increase in loading. As lower limb exoskeletons typically have some mass at both the pelvis and the thigh, the response of a user to a complete exoskeleton is likely to include characteristics of both locations in combination.

Hip exoskeleton designers should consider modifying the control approaches at the hip and knee joints to reduce the effects on ankle moments, as increased ankle moments are observed as a general response to added mass. We also observed that, under pelvis loading, middle-aged adults had a preferred adaptive strategy to avoid increased hip flexion at heel strike and increased knee flexion during weight acceptance. Under thigh loading, a general strategy of limb stiffening was observed in the case of middle-aged adults. Without exoskeleton modification, these responses could cause increased joint reaction forces and loading rates. A low magnitude of mass on the thigh seems to be well tolerated by healthy adults. However, it is also important to consider the toe-clearance requirements, and the steeper loading rates that have been shown to be associated with thigh loading, in this study. In the case of thigh loading, ideally the hip exoskeleton should also be capable of providing an additional external hip extension moment during contralateral toe-off, to meet the demands of propelling the added mass forward. Overall, as the age of the user may impact their response to an exoskeleton, designers should aim to include sensors to directly monitor user response and adaptive control approaches that account for these differences.

## Figures and Tables

**Figure 1 sensors-22-06154-f001:**
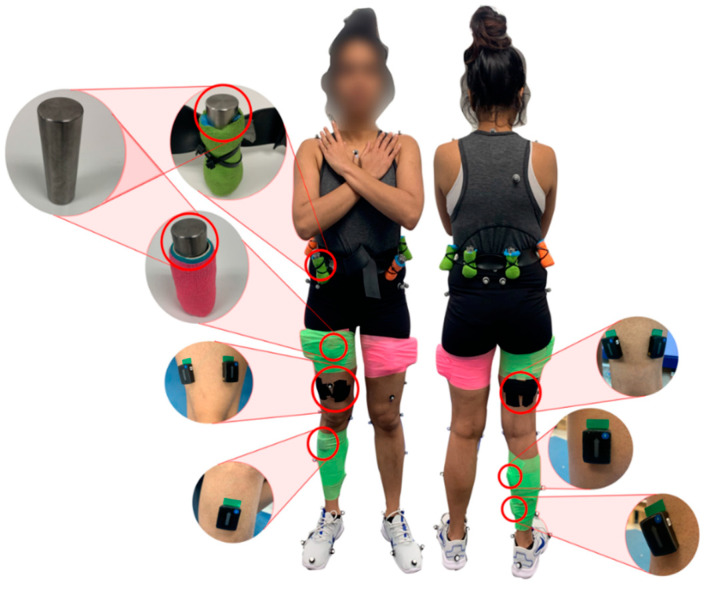
Experimental setup consisting of tungsten mass, mass holders, surface EMG sensors and retroreflective motion capture markers. Markers were placed symmetrically on the iliac crest, posterior superior iliac spine, anterior superior iliac spine, femur lateral epicondyle, femur medial epicondyle, fibula apex of lateral malleolus, tibia apex of medial malleolus, posterior surface of calcaneus, head of second metatarsal bone, head of first metatarsal bone, head of fifth metatarsal bone and acromion. A marker was also placed on the 7th cervical vertebrae and the sternum jugular notch. Additional tracking markers were placed on leg segments, with two on each thigh and one on each shank.

**Figure 2 sensors-22-06154-f002:**
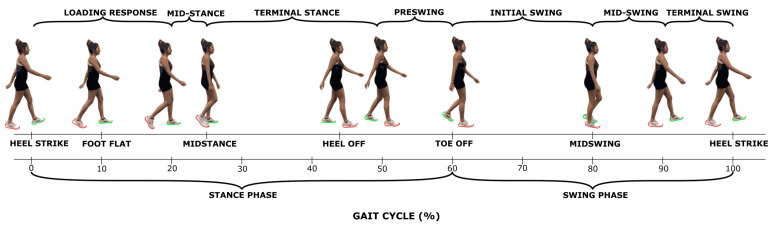
Representation of a gait cycle of the foot highlighted in green, along with the classically considered gait events and phases.

**Figure 3 sensors-22-06154-f003:**
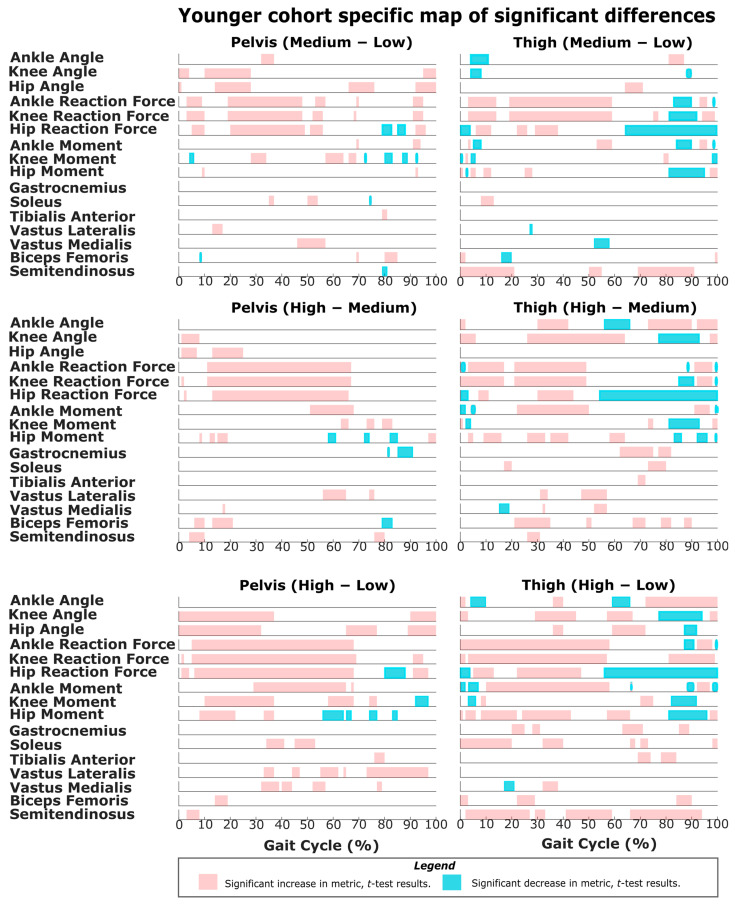
Map of all the SPM results for younger adults. Column 1 contains the results from the *t*-test for a full-factorial comparison of the pelvis masses. Column 2 contains the results from the *t*-test for a full-factorial comparison of the thigh masses. An increase in the metric (higher weight causes more ‘+’ve change) is highlighted in red, and a decrease in the metric (higher weight causes more ‘−’ve change) is highlighted in blue. Results from row 3 are also represented in Figure 5, Figure 7 and Figure 9 to indicate regions of significant differences.

**Figure 4 sensors-22-06154-f004:**
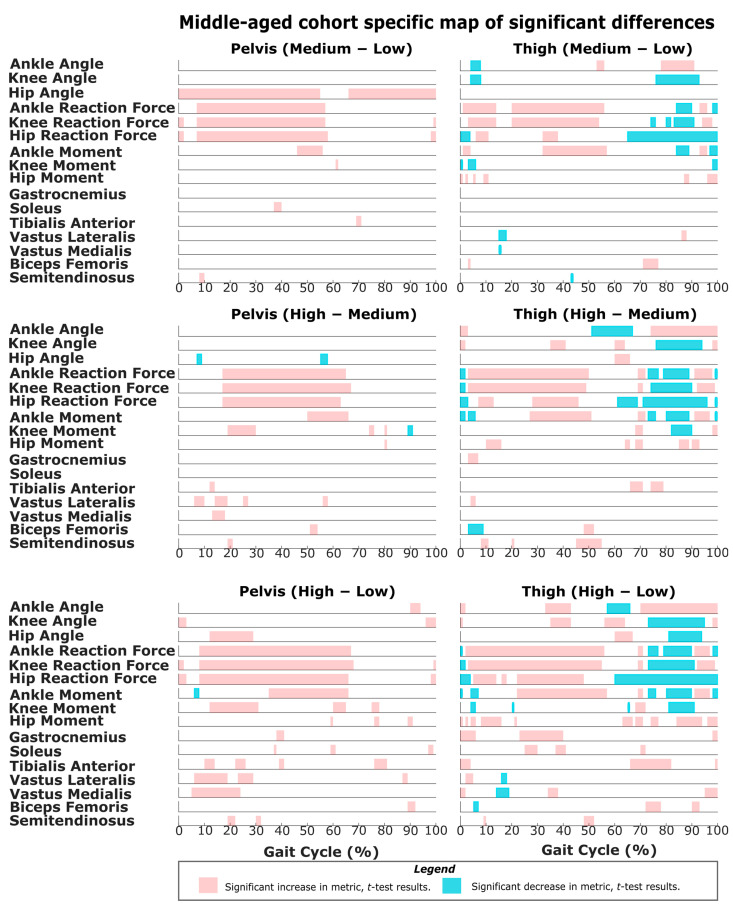
Map of all the SPM results for middle-aged adults. Column 1 contains the results from the *t*-test for a full-factorial comparison of the pelvis masses. Column 2 contains the results from the *t*-test for a full-factorial comparison of the thigh masses. An increase in the metric (higher weight causes more ‘+’ve change) is highlighted in red, and a decrease in the metric (higher weight causes more ‘−’ve change) is highlighted in blue. Results from row 3 are also represented in Figure 6, Figure 8 and Figure 10 to indicate regions of significant differences.

**Figure 5 sensors-22-06154-f005:**
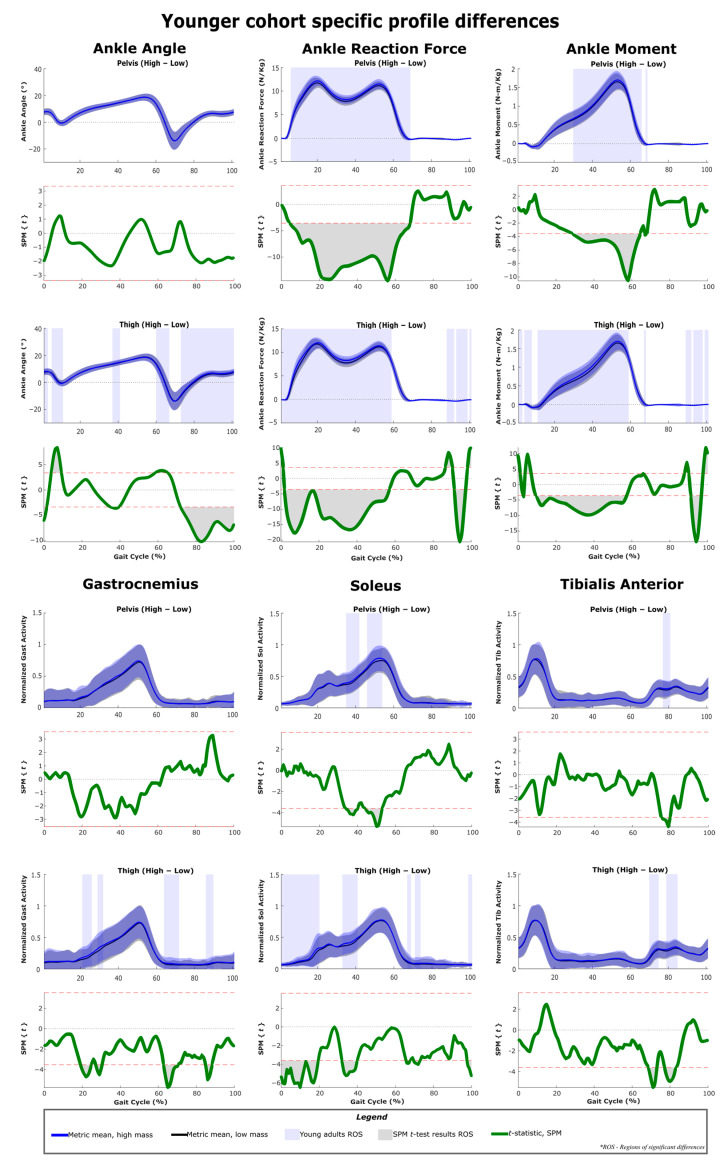
SPM *t*-test results and profile plots of the metrics, comparing low mass to high mass in the pelvis and thigh segments for participants in the younger cohort. Joint parameters are primarily related to the ankle joint. The *t*-statistic is qualitatively identical to the effect size and can be used as an indicator of practical significance.

**Figure 6 sensors-22-06154-f006:**
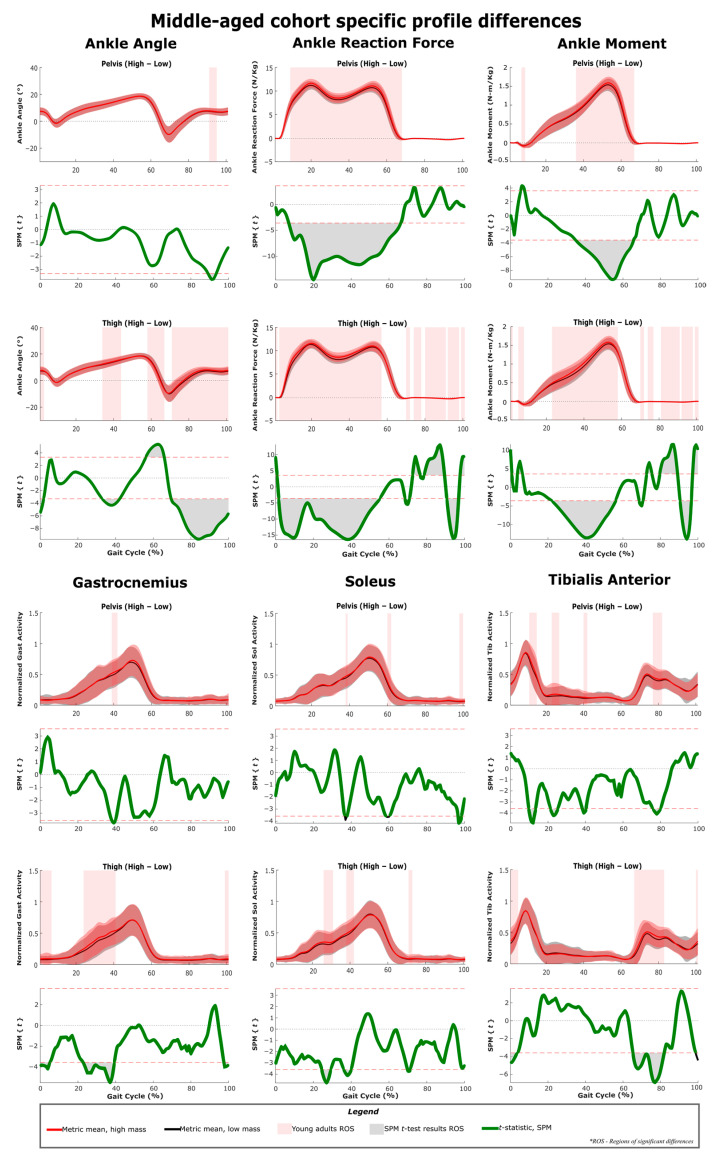
SPM *t*-test results and profile plots of the metrics, comparing low mass to high mass in the pelvis and thigh segments for participants in the middle-aged cohort. Joint parameters are primarily related to the ankle joint. The *t*-statistic is qualitatively identical to the effect size and can be used as an indicator of practical significance.

**Figure 7 sensors-22-06154-f007:**
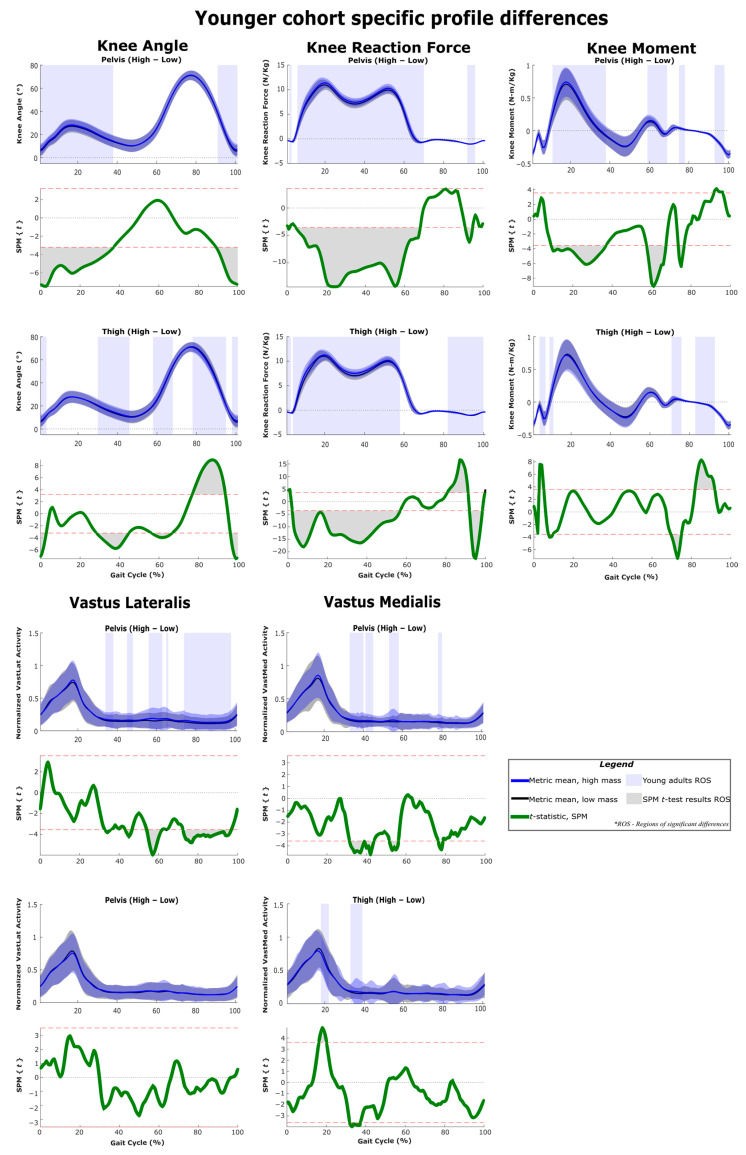
SPM *t*-test results and profile plots of the metrics, comparing low mass to high mass in the pelvis and thigh segments for participants in the younger cohort. Joint parameters are primarily related to the knee joint. The *t*-statistic is qualitatively identical to the effect size and can be used as an indicator of practical significance.

**Figure 8 sensors-22-06154-f008:**
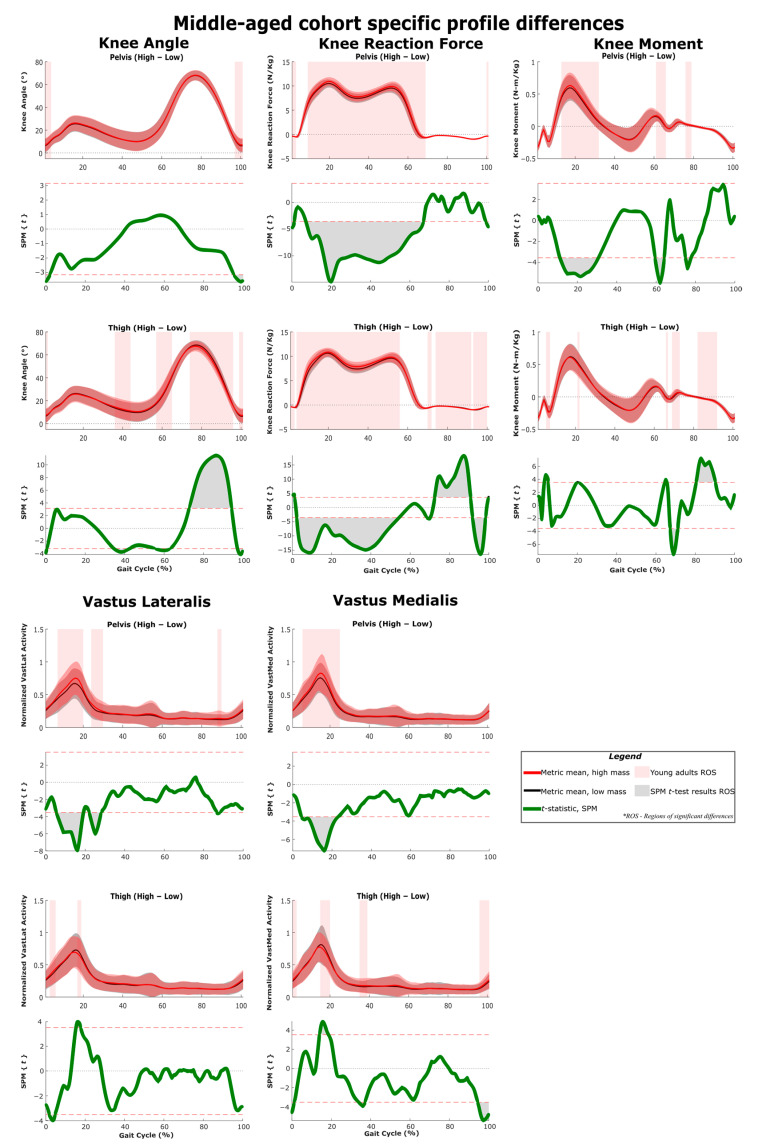
SPM *t*-test results and profile plots of the metrics, comparing low mass to high mass in the pelvis and thigh segments for participants in the middle-aged cohort. Joint parameters are primarily related to the knee joint. The *t*-statistic is qualitatively identical to the effect size and can be used as an indicator of practical significance.

**Figure 9 sensors-22-06154-f009:**
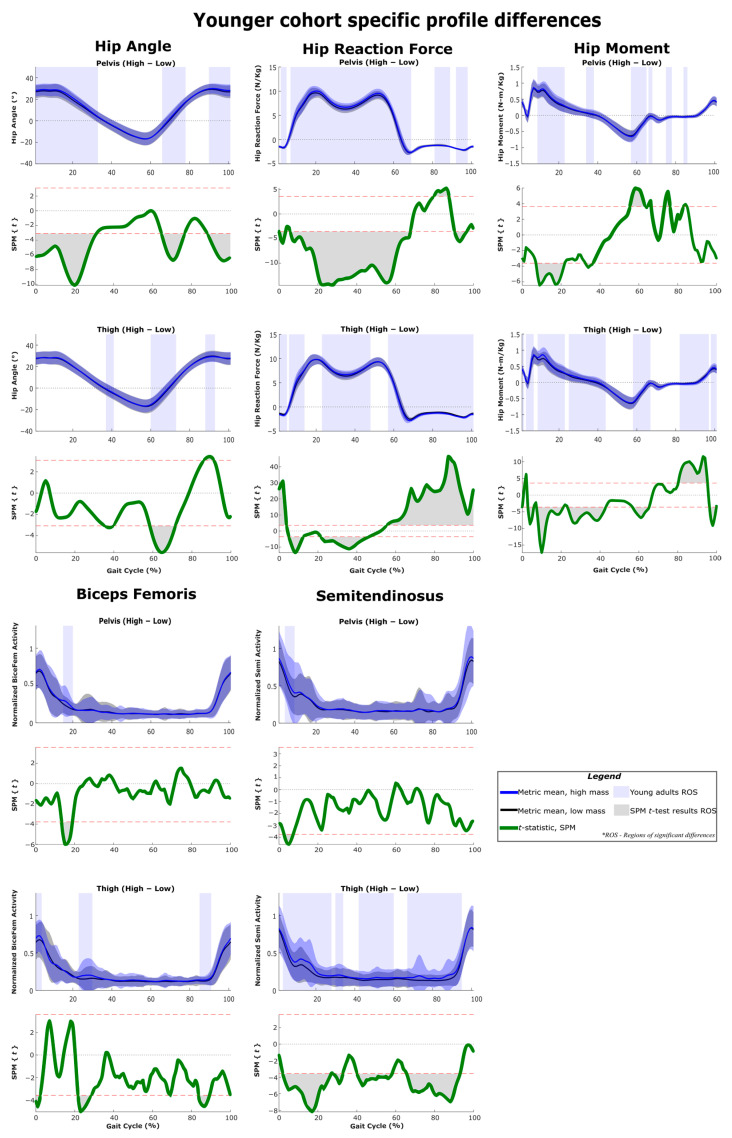
SPM *t*-test results and profile plots of the metrics, comparing low mass to high mass in the pelvis and thigh segments for participants in the younger cohort. Joint parameters are primarily related to the hip joint. The *t*-statistic is qualitatively identical to the effect size and can be used as an indicator of practical significance.

**Figure 10 sensors-22-06154-f010:**
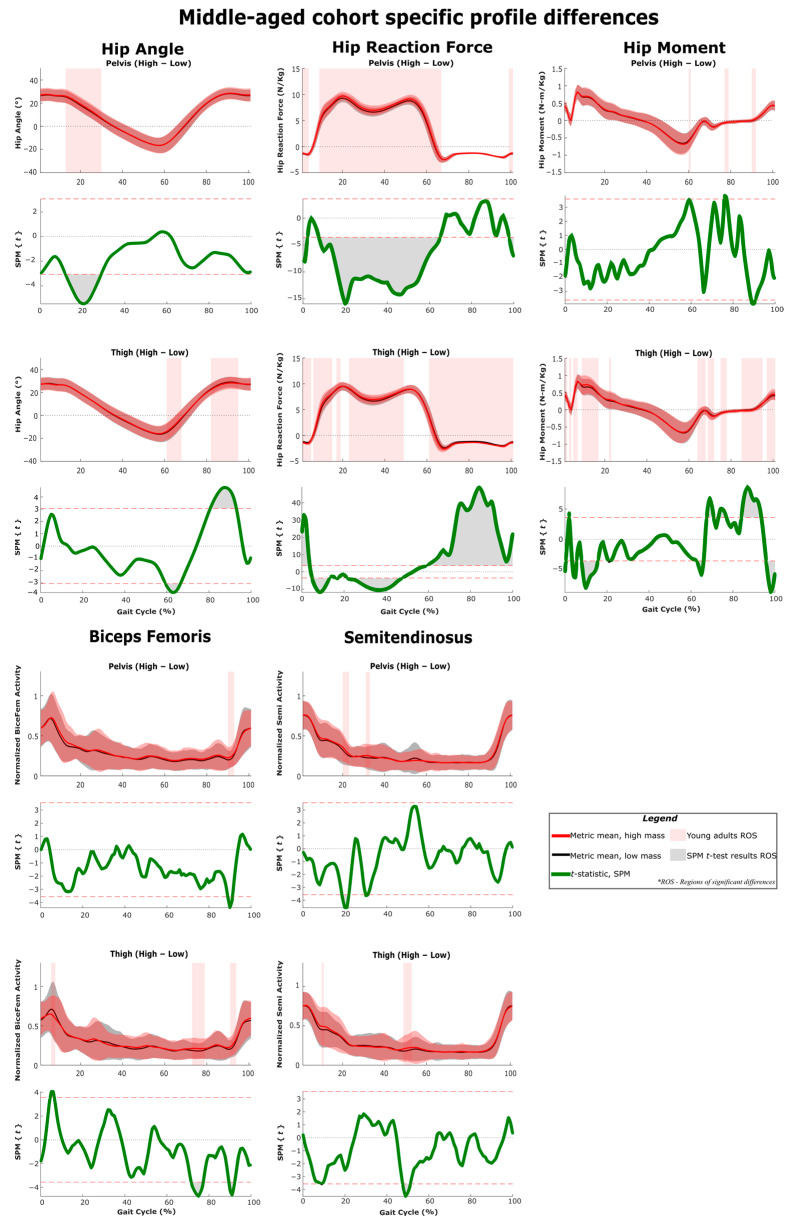
SPM *t*-test results and profile plots of the metrics, comparing low mass to high mass in the pelvis and thigh segments for participants in the middle-aged cohort. Joint parameters are primarily related to the hip joint. The *t*-statistic is qualitatively identical to the effect size and can be used as an indicator of practical significance.

**Figure 11 sensors-22-06154-f011:**
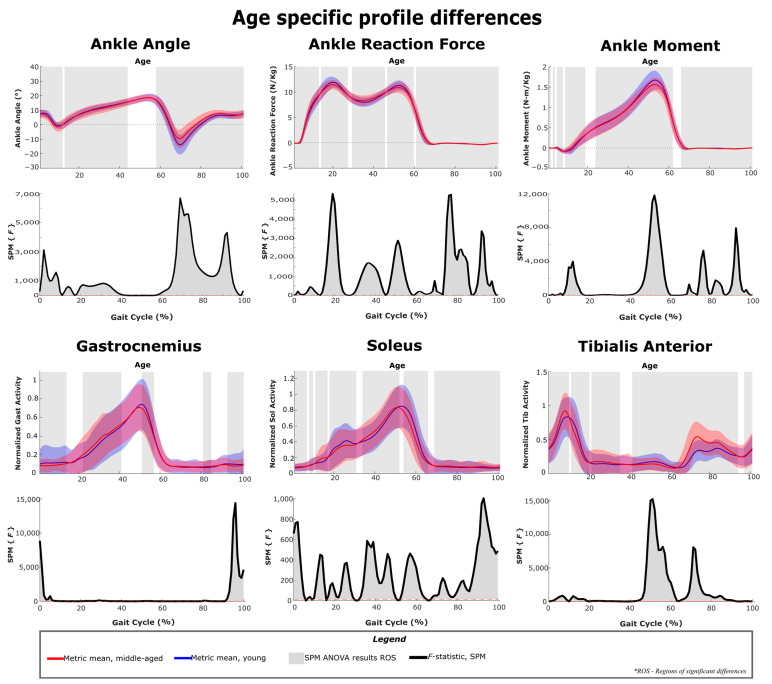
SPM ANOVA results and age-specific differences between the younger and middle-aged cohorts. Joint parameters are primarily related to the ankle joint. The *F*-statistic is qualitatively identical to the effect size and can be used as an indicator of practical significance.

**Figure 12 sensors-22-06154-f012:**
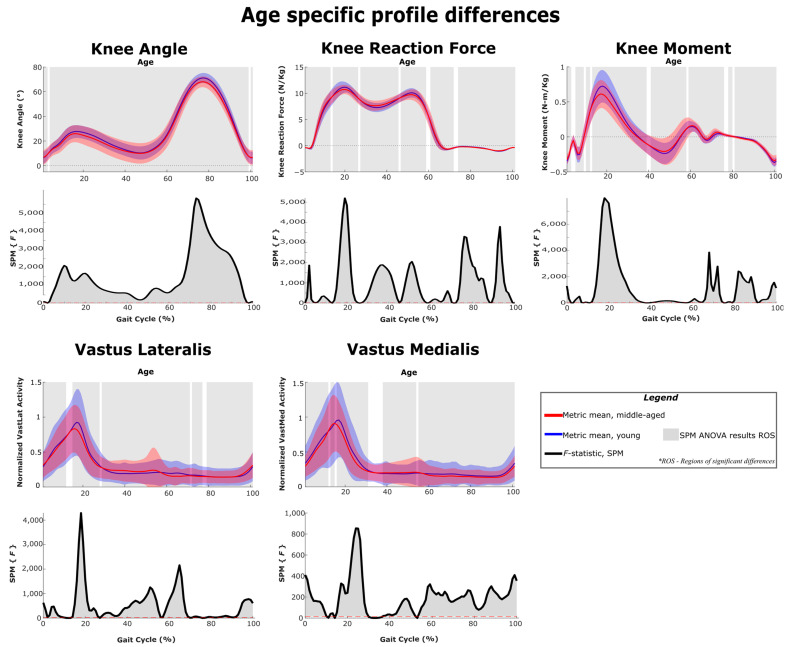
SPM ANOVA results and age-specific differences between the younger and middle-aged cohorts. Joint parameters are primarily related to the knee joint. The *F*-statistic is qualitatively identical to the effect size and can be used as an indicator of practical significance.

**Figure 13 sensors-22-06154-f013:**
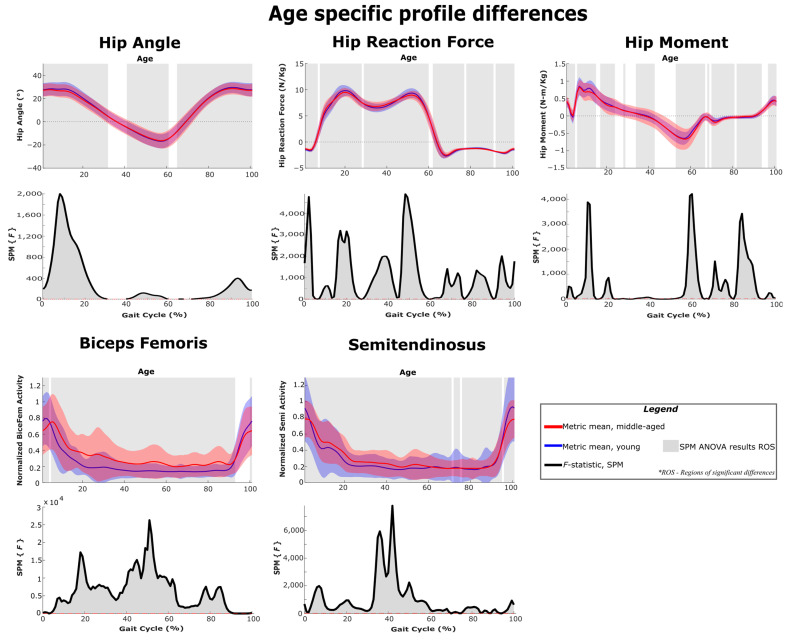
SPM ANOVA results and age-specific differences between the younger and middle-aged cohorts. Joint parameters are primarily related to the hip joint. The *F*-statistic is qualitatively identical to the effect size and can be used as an indicator of practical significance.

**Table 1 sensors-22-06154-t001:** The 9 loaded conditions represented full-factorial combinations of bilateral added mass at the thighs and the pelvis. The combinations consisted of a low (+3.6 lb/+1.6 kg), medium (+7.2 lb/+3.3 kg) and high (+10.8 lb/+4.9 kg) mass amounts at each segment.

Condition	Thigh Mass	Pelvis Mass
1	Low	Low
2	Low	Medium
3	Low	High
4	Medium	Low
5	Medium	Medium
6	Medium	High
7	High	Low
8	High	Medium
9	High	High

## Data Availability

Not applicable.

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
