# Peer review of "How Does Added Mass Affect the Gait of Middle-Aged Adults? An Assessment Using Statistical Parametric Mapping"

_sensors, 2022, doi:10.3390/s22166154_

Round 1

Reviewer 1 Report

Manuscript# sensors-1824387

Title: How does Added Mass Affect the Gait of Middle-Aged Adults? An Assessment Using Statistical Parametric Mapping

General Comments

The goal of this study was to characterize the effects of added mass on the gait parameters of middle-aged adults across the whole gait cycle and to provide insights into mass distribution patterns that would minimize the effects of added mass on middle-aged adult’s natural gait. The authors hypothesized that younger and middle-aged adults will have different responses to the added mass representative of an exoskeleton, that this will include altered muscle activation patterns between the groups and that loading at the pelvis and thigh will have different characteristic impacts on gait parameters with pelvis loading having greater impact on stance and thigh loading having greater impact on swing. While the topic is interesting and relevant and the manuscript is well-written, there are several major aspects requiring revision:

Comments

1.          As described above, the authors address several research questions:

a.     What are the effects of added mass on the gait parameters of middle-aged adults across the whole gait cycle?

b.     Which mass distribution patterns would minimize the effects of added mass on middle-aged adult’s natural gait?

c.     Does the response to the added mass representative of an exoskeleton differ between younger and middle-aged adults?

a. described by SPM (Figure 4).

b. described by SPM although the question of minimization cannot be addressed, hence the research question must be rephrased.

c. not described in the current manuscript as this would require testing for the interaction between the factors added mass and age.

The manuscript is very long and difficult to follow because it is unclear how each of these questions is addressed. The authors should consider splitting the manuscript into two or three manuscripts accordingly. This would facilitate a more focused discussion of each of these relevant topics.

2.          Some of the figures contain duplicate results, e.g., Figure 3 and 4 and Figures 5–10. One of the figure sets should be presented in as supplementary material.

3.          One of the rationales for using SPM stated by the authors is that differences across the entire gait cycle can be described. However, SPM does not consider possible differences in timing between gait and kinematic events and such differences in timing would not be detected. This aspect must be included in the limitations section.

4.          The sample size is limited. Please include information on sample size estimations.

5.          The authors focus on statistical differences. However, some of the differences are very small in magnitude. The authors should report effect sizes and discuss the functional and clinical relevance of their results.

Specific comments

6.          Abstract. It is unclear what the term “gait performance” represents.

7.          Abstract. “…did not adapt the kinematic changes…” This sentence is unclear. Please rephrase.

8.          Abstract. “…generally stiffened the leg…” This is an overinterpretation of the data as no data on leg stiffness is presented (the data to calculate this should be available).

9.          Please report the BMI of both groups.

10.       Line 102. Please either describe the marker set and define the joint coordinate systems or provide a reference.

11.       Line 106. Considering that masses were added to the thigh and pelvis it is surprising that the authors did not measure muscle activity of any hip muscles.

12.       Line 113: acclimate -> acclimatize

13.       Line 115: “Bilateral added mass…” Please rephrase, e.g., Additional masses were added bilaterally…”

14.       Were both groups experienced in treadmill walking? If not, then any differences between groups may be related to treadmill walking experience rather than to age.

15.       Figure 1. Did the pelvis masses interfere with arm swing?

16.       Line 146: Joint coordinate systems must be defined.

17.       Line 155: How many steps were analyzed? How were different steps entered into the analysis?

18.       Line 164: Description of testing for the interaction effect (i.e., the main research question) is missing.

19.       Figure 3 and 4. According to the figure caption, an increase in the metric (higher weight causes more ‘+’ve change) is highlighted in red, and a decrease in the metric (higher weight causes more ‘-’ve change) is highlighted in blue. The graphs should be labeled accordingly, i.e., high–low and not vice versa.

Reviewer 2 Report

In this study, authors investigated the kinematic and the activity of seven lower limb muscles in younger and middle-aged participants during walking, when different weights were added at the thigh and the pelvis. The study was performed to investigate how the different distribution of the weights of a lower limb exoskeleton may affect the walking of a human operator.

Despite the study is well written, the purpose is clearly stated, and the statistics are properly designed, I have some major doubts on the purpose of the study.

As correctly specified by the authors in the conclusion, a lower limb exoskeleton, especially an active exoskeleton, should compensate the impact of its own added mass, otherwise I’m not expecting operators would benefit from its use. Therefore, why investigate the effect of different loads displacements during walking if the weigh of an exoskeleton should be self-compensated? May you discuss more in detail this point.

One of the effects of the aging is the reducing of the force a participant may exert. Since the added masses are the same for the two groups, I’m wondering whether the differences in kinematics and muscle activities identified between the young and the middle-aged groups could be ascribed to the different efforts the two groups may exert after a fixed weight.

Therefore, I suggest adding a measure of fatigue, such as the median frequency of the EMG (as described in Mañanas  et al, 2008, DOI: 10.1007/BF02347701, or more recently in Borzelli et al. 2020, DOI: https://doi.org/10.1088/1741-2552/ab6d88.

As correctly stated in the limitations, authors added the masses to only one side of the participants. I’m wondering whether this may lead to imbalance in operators that may lead to a lower-limb stiffening, and therefore co-contraction of antagonist muscles, especially in middle-aged participants. Therefore, I suggest adding a measure of co-contraction (e.g. as the one proposed in Borzelli et al. 2020 previously suggested)

I suggest adding the producer and the model of the EMG sensors that you used in your study (in Figure 1 it looks that they were Delsys Trigno wireless sensors).

Round 2

Reviewer 1 Report

Manuscript# sensors-1824387

Title: How does Added Mass Affect the Gait of Middle-Aged Adults? An Assessment Using Statistical Parametric Mapping

General Comments

The reviewers have thoroughly revised their manuscript. The readability of the manuscript has greatly benefitted from the restructuring. However, several major points have not been adequately addressed.

Comments

1.          The primary hypothesis was that age will affect the influence of additional weight on gait and muscle activity patterns. I had already raised concern that this aspect was not formally tested as interaction between the factors age and weight was not included in the statistical parametric mapping model. The authors still only present data on the effect of only age or only weight. In their response, the authors argue that considering the significant overall age affect, they chose to qualitatively describe the difference in the effect of load between groups. The authors must provide adequate data and analyses for their conclusion claims.

2.          The authors focus their interpretation solely on statistical significance. It is particularly striking that they emphasize the results of the SPM by placing the figures above the gait parameter trajectories. Most literature will show the data and below the results of the SPM. Inspecting the gait pattern profiles, most statistically significant differences show only slight differences in gait parameter trajectories. In the first review, I raised the concern about clinical significance, which was indeed not correct as pointed out by the authors. However, I would argue that most of the observed statistically significant differences are functionally irrelevant. Specifically, I would expect that the authors provide some thresholds in parameters that must be exceeded that would necessitate active compensation by an exoskeleton. Without placing the magnitude of the observed difference in this context, the value of the current study is highly questionable.

3.          The authors have now included marker locations and a rough description of the joint coordinate system. However, they should add information on how the joint coordinate systems where defined. I assume that a standing trial was collected prior to data collection and the joint coordinate system defined there. The subject depicted in Figure 1 stands in a posture with externally rotated legs. If this was indeed the case, then the joint coordinate axes (aligned to the lab coordinate system) would not coincide with the anatomical axes as all joints of the lower extremity are externally rotated. This would lead to cross talk between angles in the anatomical planes. Moreoever, stating that the coordinate systems were defined as the standard in visual 3D is not sufficient as not all readers are familiar with their definitions. Either the coordinate system (including all joint center locations!) must be adequately described, or an appropriate reference provided.

Reviewer 2 Report

I would like to thank the authors for ammending the paper accordingly to my comments

Author Response

Thank you for the quick and well-thought-out review of our paper. We appreciate the time and effort that you dedicated to review our work. 

Round 3

Reviewer 1 Report

Title: How does Added Mass Affect the Gait of Middle- Aged Adults? An Assessment Using Statistical Parametric Mapping

General Comments
The reviewers have thoroughly re-revised their manuscript to allow data reproduction and included adequate conclusions supported by the presented data.

Author Response

We appreciate the time and effort that you have dedicated to reviewing our work. Thank you for your insightful comments. Your comments have helped us make this work stronger.